# Using a Deliberative Poll on breast cancer screening to assess and improve the decision quality of laypeople

**Manja D. Jensen** [1,2]*, **Kasper M. Hansen**[3], **Volkert Siersma**[1], **John Brodersen**[1,2]

**1** Department of Public Health, The Research Unit for General Practice and Section of General Practice, University of Copenhagen, Copenhagen, Denmark, **2** Primary Health Care Research Unit, Region Zealand, Denmark, **3** Department of Political Science, University of Copenhagen, Copenhagen, Denmark

* madj@sund.ku.dk

**Data Availability Statement:** Code and anonymised dataset files are available from the Harvard Dataverse database: https://dataverse. harvard.edu/dataverse/MKVJ (https://doi.org/10.

## Abstract

Balancing the benefits and harms of mammography screening is difficult and involves a value judgement. Screening is both a medical and a social intervention, therefore public opinion could be considered when deciding if mammography screening programmes should be implemented and continued. Opinion polls have revealed high levels of public enthusiasm for cancer screening, however, the public tends to overestimate the benefits and underestimate the harms. In the search for better public decision on mammography screening, this study investigated the quality of public opinion arising from a Deliberative Poll. In a Deliberative Poll a representative group of people is brought together to deliberate with each other and with experts based on specific information. Before, during and after the process, the participants' opinions are assessed. In our Deliberative Poll a representative sample of the Danish population aged between 18 and 70 participated. They studied an online video and took part in five hours of intense online deliberation. We used survey data at four timepoints during the study, from recruitment to one month after the poll, to estimate the quality of decisions by the following outcomes: 1) Knowledge; 2) Ability to form opinions; 3) Opinion stability, and 4) Opinion consistency. The proportion of participants with a high level of knowledge increased from 1% at recruitment to 56% after receiving video information. More people formed an opinion regarding the effectiveness of the screening programme (12%), the economy of the programme (27%), and the ethical dilemmas of screening (10%) due to the process of information and deliberation. For 11 out of 14 opinion items, the within-item correlations between the first two inquiry time points were smaller than the correlations between later timepoints. This indicates increased opinion stability. The correlations between three pairs of opinion items deemed theoretically related a priori all increased, indicating increased opinion consistency. Overall, the combined process of online information and deliberation increased opinion quality about mammography screening by increasing knowledge and the ability to form stable and consistent opinions.

7910/DVN/ZQHO5W, https://doi.org/10.7910/DVN/WFJIUV, https://doi.org/10.7910/DVN/SSNZJL, https://doi.org/10.7910/DVN/DYFUS9).

**Funding:** The project was funded by: Region Zealand, Den forskningsfremmende pulje (JB, www.regionsjaelland.dk), Region Zealand, PhD grant (MDJ, www.regionsjaelland.dk), Helsefonden (MDJ, 17-B-0238, www.helsefonden.dk); Fonden for almen praksis (MDJ, A1525, www.laeger.dk/fonden-for-almen-praksis); Poul og Agnes Friis fond (MDJ, 81008-001); Lilly og Herbert Hansens fond (MDJ, 051), and A.P Møllers lægefond (MDJ, 18-L-0021, https://www.apmollerfonde.dk/). The funders had no role in study design, data collection and analysis, decision to publish, or preparation of the manuscript.

**Competing interests:** The authors have declared that no competing interests exist.

## Introduction

Cancer screening is intuitively appealing and is a standard in modern healthcare. It can benefit the population reducing mortality, morbidity and incidence of disease by identifying pre-symptomatic, localised stage cancer or precursors of cancer [1]. However, screening can also lead to unintended harms [2]. The most profound one, overdiagnosis, occurs in cancer screening when indolent cancers or harmless precursors are identified that were never going to cause harm in a person's remaining lifetime [3]. The harm from overdiagnosis occurs because it is not (yet) possible to distinguish the cancers (and precursors) that do not progress from those that do. They are all, therefore, treated as if they would progress to severe illness. In mammography screening, women treated for a breast cancer (or precursors) that were never going to cause any harm, experience the harms from treatment but none of the benefits [4].

In contrast to most clinical situations, where people respond to a symptom by consulting their doctor, screening is initiated by health authorities and is aimed at the asymptomatic population. Following health authorities' recommendations, it is a fundamental premise that, on average, people are better off participating in screening than not [5]. As the benefits and harms are not measured in the same way, and do not necessarily affect the same people, it is difficult to balance them, and inherent in this process is a value judgement [6]. In addition, screening is not only a medical intervention but also a social intervention posing ethical, legal and social dilemmas and considerations [7]. In this light, it has been argued that the opinions of laypeople should be considered in decisions about screening programmes.

A commonly used way of consulting the public about policy issues is through traditional opinion polls. Findings from such studies reveal widespread enthusiasm with nearly 90% of adult populations in the US and the UK agreeing that "screening is almost always a good idea" [8,9]. In addition, high participation rates in screening programmes could also be argued to reflect support for them. However, both measures require cautious interpretation. Widespread enthusiasm could reflect pseudo-opinions (opinions on a topic about which respondents have no knowledge) [10], or simply top-of-mind opinions based on one-sided information that favours screening [11–13], and limited public awareness and understanding of overdiagnosis [14,15]. Despite the increased focus on providing quality information to aid decision making, a study has shown that the decision to attend mammography screening was not based on the information provided in the invitation letter and leaflet [16]. Participation in screening can reflect social norms, perceived moral obligations and responsibilization (the emphasis of responsibility placed on individual women to contribute to their health within society) rather than support for and understanding of the screening programme [17,18].

In order to determine *informed* laypeople's *considered* opinions about mammography screening, we conducted a Deliberative Poll® [19], originated by Fishkin [20], in Denmark in September 2020. Our Deliberative Poll involved informing a representative sample of people about mammography screening through an online video as well as intense deliberation at an online citizens' assembly. Deliberative Polls combine two normative democratic ideals, representativeness and deliberation, in one method and represent public opinion as it would be if all citizens had had the opportunity to deliberate based on balanced information.

The aim of the study was twofold: first, to design and conduct an online Deliberative Poll with a representative sample of the Danish population engaging in online deliberation. Second, to analyse how information and deliberation affected the participants' knowledge and opinion formation.

## Applying citizens' involvement in mammography screening— Communication, assessment of opinions and deliberation

Our Deliberative Poll design on mammography screening had three parts: 1) a video communication about mammography screening; 2) assessment of knowledge and opinions using questionnaires at four timepoints and, 3) a citizens' assembly with a representative sample of the population, neutral moderators, and three experts available to answer questions. The following sections describe the knowledge basis for all three parts as well as the design and outcome hypotheses of the study.

**Communication about mammography screening.**   Informing the public about mammography screening has been subject to intense criticism [21]. Debate concerns which facts should be presented, what constitutes best available evidence, in which format the information should be communicated and if misconceptions should be addressed [22]. Communication about screening is linked to risk and uncertainty: the risk of getting and dying of a disease and the risk of being harmed by the procedures in place to lower the risks. Incorporating personalised risk estimates within communication interventions for screening programmes has been shown to enhance informed choices [23]. Solid risk data about mammography screening is outdated as randomised controlled trials were conducted years ago (mostly in the 1970s and 1980s), with less optimal breast cancer treatment available at the time and risk estimates that are often uncertain because of non-randomised data collection [4]. Risk information is known to be difficult to comprehend. Numerous studies have documented statistical illiteracy among laypeople and physicians [24–26]. No standard exists on how to communicate risk most effectively. Metanalyses suggest that numbers should be presented as natural frequencies [27,28], however, the most suitable format depends on the nature of the task [29,30]. Literature suggests that icon arrays may aid understanding and should be formatted consistently with a shared common reference class [29,31]. In addition, recognising uncertainty and presenting screening as a choice are perceived as important by laypeople [32,33]. Taking all of these considerations into account, much effort was put into developing a balanced video about mammography screening for the present study.

**Assessing the opinions of laypeople about mammography screening.**   Opinions about screening, like most social concepts, are complex and multifaceted. The processes involved in opinion formation are not fully understood. Citizens' Juries about mammography screening have provided knowledge about aspects other than harms and benefits that are deemed important to women; for example, the freedom to choose and the symbolic importance of screening —the idea that society cares about women [34,35]. The psychological pathways through which participants process information about overdiagnosis in a mammography screening leaflet, and how this influences their attitudes and intention to participate, have also been studied [36]. The study revealed both cognitive and affective mediators. Knowledge from these studies was used for questionnaire development in the present study, incorporating questions related to the possible cognitive, affective and social aspects of opinions about mammography screening.

**The Deliberative Poll–Deliberation and representativeness.**   To include everyone in a discussion while also ensuring that they are motivated and can reflect on the issue at stake is a difficult task. Nevertheless, the Deliberative Poll is designed to do exactly that [37]. In a Deliberative Poll a representative group of people is brought together to deliberate with each other and with experts based on specific information. Before, during and after the process, the participants' opinions are assessed. In Deliberative Polling, representativeness is defined by compliance with three criteria: 1. Demographics; 2. Attitudes, and 3. Sample size. To support the inference that entire groups of citizens would come to the same conclusions if they had the

opportunity to deliberate on the basis of receiving the same information, Deliberative Polls have to comply with demographic and attitudinal representativeness. In addition, a sufficient sample is needed to meaningfully evaluate the representativeness as well as any statistical significance in knowledge and consistency change [37]. It is not straightforward, however, to bring together a representative sample of people to debate a complex issue like mammography screening. Certain groups may self-select or de-select, based on prior knowledge, opinion and engagement with the programme. It is just as difficult to create a free and open atmosphere for deliberation, where everyone can address their concerns and raise questions. As such, the Deliberative Polls' two design-hypotheses can be voiced: *(a)* it is possible to create representative processes on a national level where *(b)* deliberation about the issue at stake takes place.

**Judging the quality of public opinions—Considerations and hypotheses.** From a deliberative democratic standpoint, the quality of opinions arising from a Deliberative Poll can be judged according to the normative democratic ideals related to information and deliberation. If opinions that underlie the decision are informed and deliberate, the decision can be argued to be of good quality. However, information and deliberation have the potential to affect the process of opinion formation in the following three areas—in both desirable and undesirable ways.

First, by affecting knowledge. The detailed information provided to participants along with the opportunity to share knowledge in the deliberative process might bring different types of information into play, thereby increasing the pool of information available to each participant. In addition, the deliberative process allows for clarification of questions. However, as has been documented previously in screening settings, cognitive dissonance might be a barrier to increased knowledge. To be confronted with knowledge that conflicts with previously held beliefs about screening may give rise to feelings of discomfort that can lead to the rejection of new information as a way of coping with the discomfort [38–40].

Second, by affecting the ability to form reasoned and stable opinions. The more information and the more arguments the participants encounter during the process of deliberation, the less likely they are to be confronted with unknown arguments and new information later on, which could alter their opinions. Opinion change during the process is expected to occur step by step and not as a random process [41–43].

Third, by affecting opinion consistency. The process may lead participants to identify inconsistencies in their own arguments. As inconsistencies make an argument hard to follow in discussions, participants may try to avoid inconsistencies. A counterargument implying that participants become less able to form opinions, and form opinions that are less stable and less consistent, is that complex information and deliberation leave participants more in doubt and more confused.

In light of the normatively desirable and undesirable potentials of information and deliberation already mentioned, we hypothesised as follows: Hypothesis 1: Participants will increase their knowledge about mammography screening as a result of the process of video information and deliberation. Hypothesis 2: The process of information and deliberation will result in an increased ability to form opinions, which will be reflected in fewer 'don't know' answers in the questionnaires and will result in the formation of opinions that are more stable. Hypothesis 3: Participants will have more clarity in their views, which will be reflected in increased consistency between items deemed theoretically related. By testing these hypotheses, this study aims to analyse the quality of public opinion about mammography screening: are participants able to form informed and reasoned opinions on which to base their recommendation?

## The Deliberative Poll on mammography screening

The first aim of the study was to conduct an online Deliberative Poll and thereby test the two design hypotheses: (a) it is possible to create a representative process at a national level where

(b) deliberation takes place. The following sections describe how this was accomplished in the Deliberative Poll on mammography screening. The Office of Research and Innovation at University of Copenhagen have approved the study, reference 514-0385/19-3000. According to the Committees on Health Research Ethics for the Capital Region of Denmark (De Videnskabsetiske Komiteer for Region Hovedstaden) the project does not constitute a health research project, but is considered a questionnaire-based study as defined by the "Danish Act on Research Ethics Review of Health Research Projects" Section 2. Thus, this project is not subject to notification from the Committees (Journal-no.: 21031705.) The Deliberative Poll was conducted online on Sunday 20th of September 2020 in Denmark. Using the CJC checklist [44] as a guide, the Deliberative Poll can be described as follows:

## Recruitment

Kantar Gallup Denmark ran the recruitment process using their large online panel of 50.000 citizens recruited through random sampled telephone interviews. Citizens were recruited either by their landline or their mobile phone dependent on which number was obtainable. Participants were recruited to the Deliberative Poll through a two-step quota sampling procedure which was closely supervised by MDJ and KMH. First 1,977 men and women from the total Kantar Gallup panel were e-mailed with a link to an online questionnaire (time $T_0$). These participants reflected the sociodemographics of the Danish population aged 18 to 70 by sex, age, educational level, marital status, and electoral district according to Statistics Denmark. Respondents who completed the questionnaire ($T_0$) and agreed to further contact were immediately directed to another online questionnaire ($T_1$). See Fig 1 for the recruitment flow diagram. Answers to $T_0$ revealed the distribution of responses to key items.

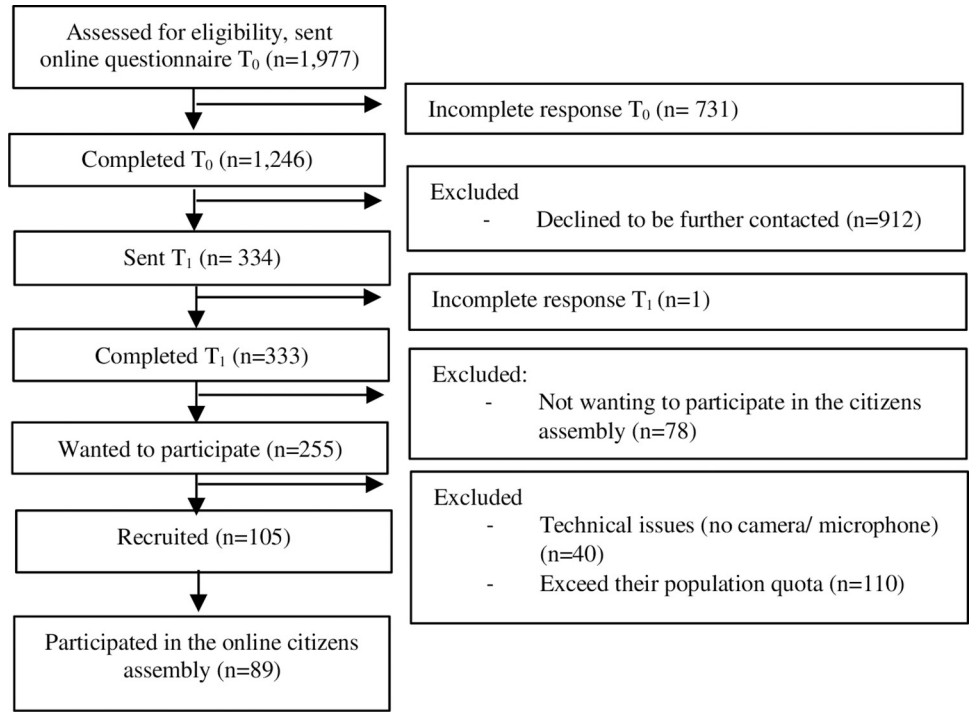

**Fig 1. Flow diagram for the recruitment to the Deliberative Poll.**

Informed consent was obtained written, online in the first questionnaire where participants was provided information about the project. Answering "Yes, I accept" and in addition entering preferred contact information into the questionnaire was considered an informed consent.

The responses to $T_0$ were used in the second step of quota sampling enabling a sample representative of the following additional parameters: family history of breast cancer, level of worry, education within healthcare, knowledge of and immediate attitude towards mammography screening. Besides enabling representativeness on key items, the second step in the quota sampling also prevented a potential bias arising if, for example, more women than men agreed to participate. A total of 89 people participated in the citizens' assembly. Participant characteristics are shown in Table 1.

The participants were assessed for sociodemographic representativeness by comparing sex, age, education, residence and marital status to the general Danish population aged 18 to 70. These characteristics were derived from 2020 register data from Statistics Denmark. The representativeness of the participants on selected key items (education within healthcare, level of worry, knowledge, initial attitude towards mammography screening and family history of breast cancer) were assessed by comparison to the pre-intervention measures from $T_0$. Overall, the target population for the present study was representative of the Danish population of 18 to 70 year-olds (Table 1). The sampling procedure succeeded in reaching a close to perfect representative sample of the Danish population within the selected age parameters, in line with design hypothesis (a) (Table 1).

## The online citizens' assembly

Participants were invited to take part in what was framed as "a citizen's assembly concerning mammography screening as part of a research project". Inclusion criteria: men and women aged 18–70, who could speak and read Danish. Exclusion criteria: no access to a computer/tablet with camera or microphone. Participants were paid 500 DKK (67 €) in compensation for their time. The assembly was held on a Sunday to minimize selection bias due to participants having to go to work.

The citizens' assembly lasted a day and participants deliberated in small group sessions led by neutral moderators, and in plenary sessions with the opportunity to ask questions of experts. See S1Table for assembly programme. The former chief of the Danish Council on Ethics chaired the assembly and was instructed to take a neutral position. Three experts, looking at the evidence from different perspectives, attended the assembly. One was a medical doctor from the Nordic Cochrane collaboration, one was an anthropologist from the patient organisation, the Danish Cancer Society, and one was a health economist experienced in mammography screening economics. Their role was to answer questions directly from the participants (not to give a general presentation) enabling participants, rather than experts, to define the problems, dilemmas and questions for discussion. The small group discussions (with eight participants in each group) were led by neutral moderators, primarily school teachers, instructed to take a neutral position and to ensure an open and equal dialogue. There was no encouragement or instruction to reach consensus in the groups.

Before the citizens' assembly all participants were encouraged to study a video informing them about mammography screening. The video (in Danish) is available at: http://bit.ly/mammografi-screening. See S2 Table for a description. In addition to the video, participants were provided with a fact sheet containing key points from the video (see S1 Fig). Sociodemographic characteristics and initial opinions were assessed online at $T_0$. Outcomes regarding knowledge, opinions and recommendations were assessed in an online questionnaire at four time points: at the end of recruitment ($T_1$), at the beginning ($T_2$) and end ($T_3$) of the citizen

**Table 1. Participant characteristics.**

| Characteristics | | Participants n = 89% | General Danish population, age 18–70 years. Data available from "Statistics Denmark" n = 3.885.797% |
|---|---|---|---|
| Sex | Men | 52 | 50 |
| | Women | 48 | 50 |
| Age group | 18–30 years old | 20 | 25 |
| | 31–40 years old | 19 | 18 |
| | 41–50 years old | 24 | 20 |
| | 51–60 years old | 20 | 20 |
| | 61–70 years old | 17 | 17 |
| Education | 7–13 years in school | 41 | 36 |
| | Vocational training | 28 | 29 |
| | Short further education | 8 | 5 |
| | Middle further education | 13 | 18 |
| | Long further education | 10 | 12 |
| Residence, electoral district in Denmark | Copenhagen | 22 | 15 |
| | Around Copenhagen | 9 | 9 |
| | Northern Zealand | 11 | 8 |
| | Bornholm | 0 | 1 |
| | Zealand | 14 | 14 |
| | Funen | 7 | 8 |
| | Southern Jutland | 11 | 12 |
| | Eastern Jutland | 17 | 14 |
| | Western Jutland | 2 | 9 |
| | Northern Jutland | 7 | 10 |
| Marital status, n (%) | Never married | 50 | 42 |
| | Married or civil partnership | 32 | 44 |
| | Divorced or separated | 14 | 12 |
| | Widowed | 4 | 2 |
| **Characteristics–Data available from the pre-intervention measures at $T_0$** | | **Participants n = 89%** | **Pre-intervention measures at $T_0$**, n varies between 1,112 and 1,290% |
| Attitude: women in Denmark should be invited to mammography screening. n = 1,290 | Strongly agree | 89 | 85 |
| | Somewhat agree | 6 | 7 |
| | Neither agree nor disagree | 2 | 3 |
| | Somewhat disagree | 1 | 1 |
| | Strongly disagree | 1 | 1 |
| | Don't know | 1 | 3 |
| Worry about breast cancer. n = 1,274 | Never | 31 | 33 |
| | Less than once a month | 53 | 46 |
| | Once a month | 8 | 9 |
| | Once a week | 2 | 2 |
| | Daily | 1 | 1 |
| | Several times daily | 2 | 1 |
| | Don't know | 3 | 8 |

(*Continued*)

**Table 1.** (Continued)

| Working/educated within the healthcare sector. n = 1,286 | Yes | 15 | 15 |
|---|---|---|---|
| | No | 85 | 85 |
| History of breast cancer in the family. n = 1,112 | Yes | 37 | 37 |
| | No | 58 | 57 |
| | Don't know | 5 | 6 |
| Conceptual knowledge • Diagnosis, correct answers • Overdiagnosis, correct answers | | 68 16 | 62 14 |

Note: The table shows the sociodemographic characteristics in the general Danish population and in participants of the Deliberative Poll.

No statistically significant differences were found between the participant sample and the general population using Chi squared test.

assembly and one month after the assembly ($T_4$). See Fig 2 for an overview of the study design including dates, order of interventions and questionnaires. For questionnaires, see S2 Fig.

The video was 20 minutes long and was shown at the beginning of the citizens' assembly. Table 2 shows the participants' self-reported compliance with the video intervention as well as participants' ability to complete an item assessing compliance. Compliance was assessed in the citizen's assembly after the screening of the video ($T_2$). 90% of the participants reported compliance in terms of preparation beforehand, as they reported having seen the video twice or more. 85% of participants were able to answer the test question correctly, indicating high compliance with the intervention.

From the participants' evaluation of the group discussions it is evident that they weighted competing arguments. 80% reported that other participants' arguments were useful in the formation of their own opinions. Participants' evaluation of group discussions is described in Table 3. See S3 Table for the full evaluation of the Deliberative Poll and for participants' evaluation of the deliberation.

95% of participants reported that group moderators made sure that all the participants had a chance to be heard (see S3 Table). In addition, 85% of participants reported that the video added new insights on mammography screening (see S3 Table).

The design of the Deliberative Poll together with the participants' evaluation of group moderators and group deliberation support design hypothesis b: an open and free deliberation

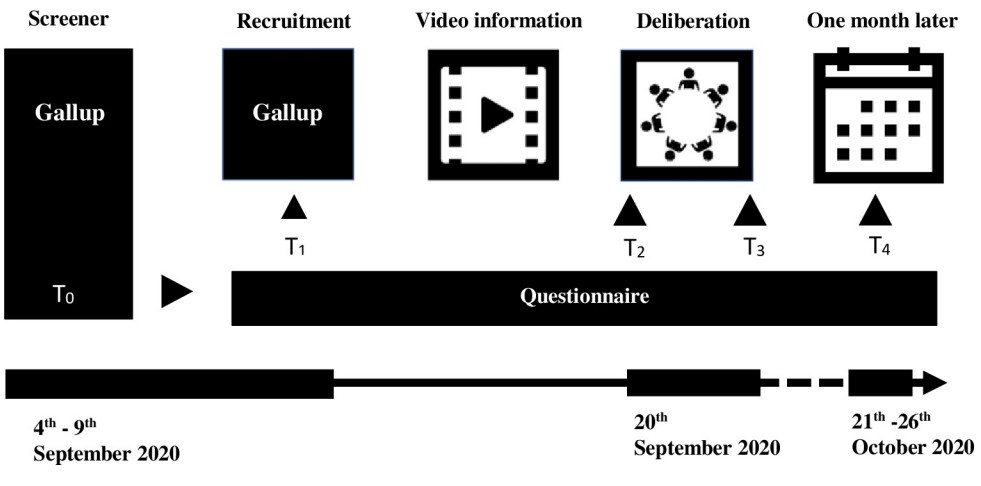

**Fig 2. Study design.**

**Table 2. Compliance with the video intervention, T$_2$ (percent).**

| Compliance | T$_2$, n = 87% |
|---|---|
| Participants indication of compliance—video watched: | |
| • Three times or more | 12 |
| • Twice | 78 |
| • Once | 10 |
| • Never | 0 |
| Test question (compliance): Correct answer to question concerning the video's visual presentation of harmful cancers | 85 |

Note: T$_2$ = inquiry time point 2 (after video information).

took place at the citizens' assembly which, together with the video information, provided the basis for learning and opinion formation in the Deliberative Poll.

## How information and deliberation affected knowledge and opinion quality in the Deliberative Poll on mammography screening

The second aim of this study was to analyse how the process of information and deliberation affected knowledge and opinion formation. It was hypothesised that: 1) Participants will increase their knowledge about mammography screening as a result of video information and deliberation. 2) The process of information and deliberation will result in an increased ability to form opinions reflected in fewer 'don't know' answers and more stable opinions. 3) Participants will have greater clarity about their views, which will be reflected in increased opinion consistency.

### 1) Knowledge

Knowledge was conceptualised as correct answers to 13 items addressing conceptual and numeric knowledge expressed with the use of a knowledge index which combined all 13 items, giving 7.69 points for each correct answer. The index ranged from 0 to 100, where 100 indicates correct answers to all questions and 0 indicates incorrect answers to all questions. Low, mid and high levels of knowledge were defined according to the knowledge index as >0, >33.33 and >66.66 respectively. The difference in proportion of high levels of knowledge and the proportion of correct answers to each of the 13 knowledge questions over the four poll inquiry time points (T$_1$-T$_4$) were assessed in linear models for binary outcomes adjusting for repeated measurements using generalised estimating equations (GEE). Fig 3 shows the percentage of participants with low, mid and high levels of knowledge. Participants increased their total knowledge markedly (Fig 3). The proportion of participants with a high level of knowledge increased from 1% at recruitment (T$_1$) to 56% after video information (T$_2$). Learning was driven by video information reflected by the increase in knowledge observed between

**Table 3. Participants' evaluation of deliberation, T$_3$.**

| Questions | T$_3$, n = 85%Agreeing |
|---|---|
| In group discussion we responded to each other's arguments | 78 |
| All opinions were listened to with the same respect | 89 |
| I got an understanding of other participants' arguments that contradicted my own | 67 |
| Other participants' arguments were useful in the formation of my own opinions | 80 |

Note: T$_3$ = inquiry time point 3 (after deliberation).

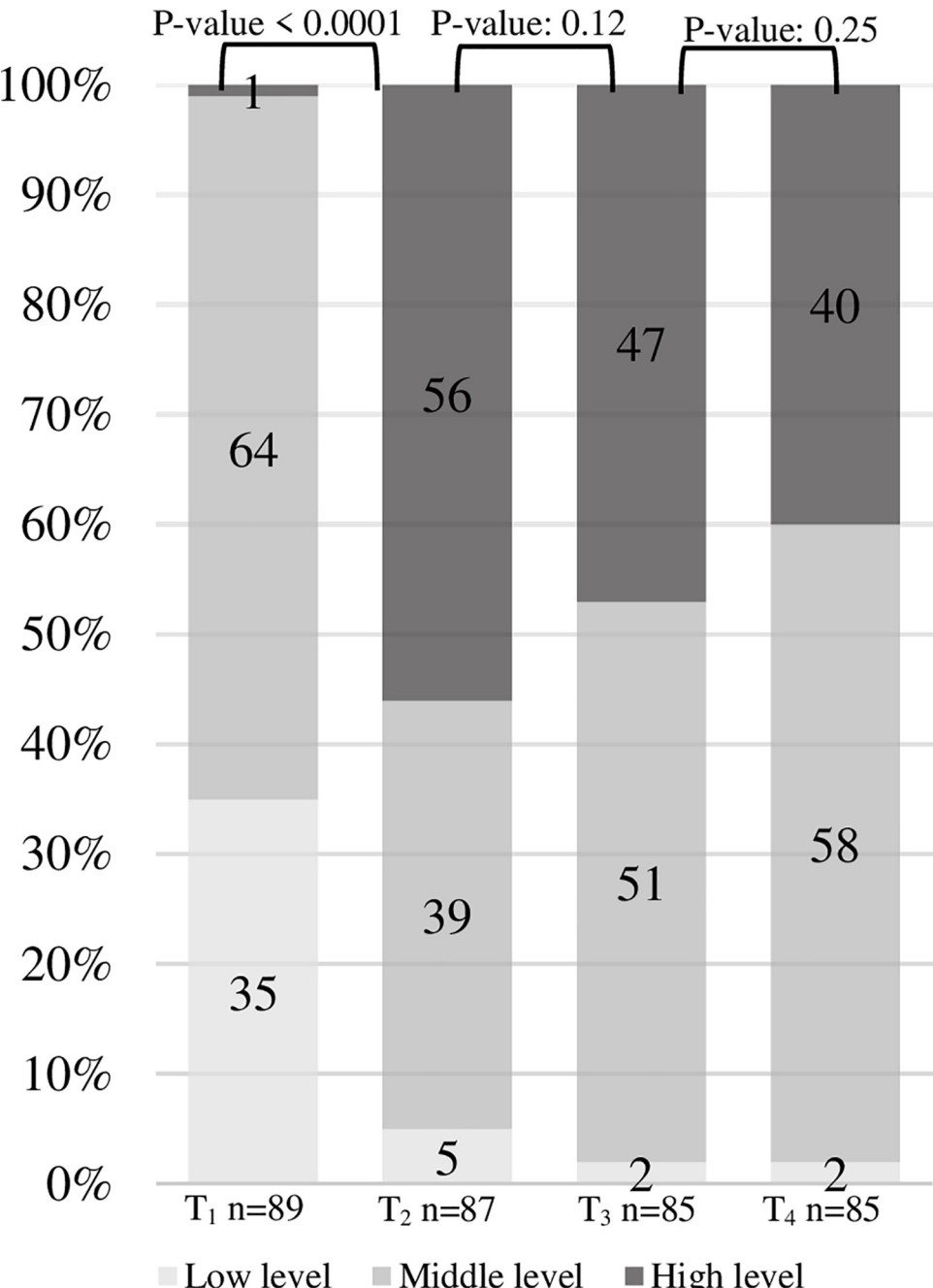

**Fig 3. Level of knowledge, %.** Note: The knowledge index combines all 13 questions giving 7.69 points for each correct answer. The index ranges from 0 to 100, where 100 indicate correct answers to all questions and 0 indicates incorrect answers to all questions. Low, mid and high levels of knowledge were defined according to the knowledge index as >0, >33,33 and >66,66 respectively. $T_1$ = inquiry time point 1 (recruitment), $T_2$ = inquiry time point 2 (after video information), $T_3$ = inquiry time point 3 (after deliberation), $T_4$ = inquiry time point 4 (one month after the citizens' assembly).

$T_1$ and $T_2$. There was no statistically significant difference in knowledge before and after the citizens' assembly, $T_2$ vs $T_3$. A decrease in knowledge is seen one month after the assembly ($T_4$), but most of the gained knowledge is persistent.

The development of knowledge expressed as a 0–100 index over the four time points, stratified for sociodemographic participant characteristics, was assessed in a linear model adjusting for repeated measurements using GEE. Learning was not restricted to specific participants as learning is seen across almost all subgroups (S4 Table). We found the largest knowledge gain in participants with the lowest level of knowledge at recruitment ($T_1$), but also noticeable knowledge gain in participants with middle levels of knowledge (S4 Table). It is impossible to conclude anything statistically significant about learning in the group of people with high levels of knowledge at recruitment ($T_1$) due to ceiling effect and because only one participant belonged to this group. Knowledge gain was also investigated for each of the 13 knowledge items separately in linear models for binary (correct vs incorrect) outcomes accounting for repeated measurements using GEE. The combined process of information and deliberation was followed by a statistically significant increase in knowledge on most items (11 out of 13, S5 Table).

To assess the occurrence of selective learning, participants were divided according to their initial recommendation about mammography screening. Participants were also divided according to decisiveness—undecided participants answering 'don't know' vs. decided participants giving any other answer. The differences in development in the knowledge index and the proportion of correct answers to each of the 13 knowledge questions over the four inquiry time points between these groups were assessed in linear models using GEE. There was no indication of selective learning (Table 4).: there was no statistically significant difference in learning between the participants with different initial opinions ("continue" vs "expand"). As the initial opinion of "discontinue" was only shared by two participants no statistical analysis was conducted comparing this group to the others. On two knowledge items there was a statistically significant difference in learning between the decided and undecided, pointing, however, in opposite directions, which may be a spurious result due to multiple testing.

These results support our first hypothesis: participants increased their knowledge about mammography screening.

## 2) Ability to form opinions and opinion stability

Participants' ability to form opinions was conceptualised as their ability to answer opinion questions with anything other than 'don't know'. The ability to form opinions for each of the 14 items over the four poll inquiry time points was assessed in linear models for binary outcomes using GEE. Statistically significantly more people formed an opinion regarding effect size (12%), costs (27%) and ethical dilemmas in screening (10%), as a drop in 'don't know' answers was seen comparing responses at recruitment ($T_1$) to responses given after deliberation ($T_3$) (Fig 4 and S6 Table). These results support our second hypothesis: the process of information and deliberation increased participants' ability to form opinions.

Question wording related to the item "Balance": Below are two statements. You can think of them as a discussion between two people, A and B. We will ask you to answer if you are most in agreement with A or most in agreement with B. Even if you do not totally agree with any of the positions, we will ask you to answer which position comes closest to your own viewpoint. How do you consider the balance between benefits and harms in mammography screening? A says: The harm carries most weight. Too many women are overdiagnosed and overtreated and getting false alarms compared to the women prevented from dying of breast cancer. B says: The benefit carries most weight. The number of women prevented from dying of breast cancer counterbalances the number of women who are overdiagnosed and overtreated and those getting false alarms. $T_1$ = inquiry time point 1 (recruitment), $T_2$ = inquiry time point 2 (after video information), $T_3$ = inquiry time point 3 (after deliberation), $T_4$ = inquiry time point 4 (one month after the citizens' assembly).

**Table 4. Learning ($T_1$-$T_3$) divided according to initial recommendation and decisiveness.**

| Learning from $T_1$ to $T_3$ | Recommendation about mammography screening in Denmark ($T_1$) | | | | Decisive regarding recommendation ($T_1$); Discontinue continue, and expand put together. n = 84 | Difference in learning between 'continue' and 'expand' (p value) | Difference in learning between decisive and undecisive (= 'don't know') (p value) |
|---|---|---|---|---|---|---|---|
| | Discontinue n = 2 | Continue n = 17 | Expand n = 65 | Don't know n = 5 | | | |
| Learning = Difference in mean total knowledge score (index) $T_1$ –$T_3$ | 30.77 | 24.04 | 26.80 | 24.62 | 26.35 | -1.61 (0.781) | -1,73 (0.745) |
| Learning = Difference (pp) in correct answers between $T_1$ and $T_3$ | PP (% correct $T_3$—$T_1$) | PP (% correct $T_3$—$T_1$) | PP (% correct $T_3$—$T_1$) | P (% correct $T_3$—$T_1$) | PP (% correct $T_3$—$T_1$) | | |
| **Knowledge, conceptual** | | | | | | | |
| Screening is for women without symptoms | 0 (100–100) | -7 (75–82) | 19 (82–63) | 60 (100–40) | 13 (81–68) | -26 (0.086) | 47 (0.041)* |
| Not all breast cancers cause illness | 0 (100–100) | 29 (94–65) | 41 (92–51) | 20 (100–80) | 38 (93–55) | -12 (0.435) | -18 (0.346) |
| Screening reduces breast cancer deaths | 50 (100–50) | 0 (94–94) | -8 (89–97) | -40 (60–100) | -5 (90–95) | 8 (0.398) | -35 (0.118) |
| Screening increases breast cancer diagnoses | -50 (50–100) | 11 (87–76) | 24 (92–68) | 60 (100–40) | 20 (90–70) | -13 (0.399) | 40 (0.077) |
| Screening leads to some women getting unnecessary treatment | 50 (100–50) | 82 (100–18) | 75 (89–14) | 80 (100–20) | 76 (91–15) | 7 (0.485) | 4 (0.819) |
| The meaning of false positive results | 0 (50–50) | 15 (62–47) | 24 (63–39) | 0 (40–40) | 22 (62–40) | -9 (0.588) | -22 (0.587) |
| Screening will not find every breast cancer | 50 (100–50) | 29 (94–65) | 32 (90–58) | 20 (80–60) | 32 (91–59) | -3 (0.829) | -12 (0.532) |
| Screening may result in prolonged life as a patient | 50 (50–0) | 53 (94–41) | 67 (85–18) | 60 (60–0) | 64 (86–22) | -14 (0.303) | -4 (0.873) |
| Benefit evaluation–reduced mortality | 50 (100–50) | 22 (81–59) | 20 (71–51) | -20 (60–80) | 21 (73–52) | 2 (0.884) | -41 (0.030)* |
| **Knowledge, numerical**** | | | | | | | |
| Breast cancer mortality without mammography screening | 50 (100–50) | 14 (31–17) | 5 (17–12) | 0 (20–20) | 8 (21–13) | 9 (0.607) | -8 (0.777) |
| Breast cancer mortality with mammography screening | 50 (50–0) | 31 (31–0) | 9 (29–20) | 20 (20–0) | 15 (30–15) | 22 (0.110) | 5 (0.774) |
| Overdiagnosis | 50 (50–0) | 25 (25–0) | 7 (19–12) | 40 (40–0) | 12 (21–9) | 18 (0.157) | 28 (0.212) |
| False positives | 50 (50–0) | 19 (25–6) | 28 (35–7) | 20 (20–0) | 27 (34–7) | -9 (0.554) | -7 (0.726) |

Note: The table shows learning divided according to initial recommendation and decisiveness regarding mammography screening at $T_1$. $T_1$ = inquiry time point 1 (recruitment), $T_3$ = inquiry time point 3 (after deliberation).

** Numbers accepted as correct: Breast cancer mortality without mammography screening: 8-14/1000 women in a 20-year period, Breast cancer mortality with mammography screening: 8-14/1000 women in a 20-year period offered screening every second year, Overdiagnosis: 8-14/1000 women in a 20-year period offered screening every second year. False positive: 100-200/1000 women in a 20-year period offered screening every second year.

Stability was operationalised as correlations *within* opinion items at *different* time points. For 11 out of 14 opinion items, the within item correlations between the time of recruitment ($T_1$) and the time after video information ($T_2$) were smaller than the correlations between the time after video information ($T_2$) and the time after deliberation ($T_3$) (S3 Fig), indicating increased opinion stability as a result of the process. In addition, for all items the mean of sequential order correlations was larger than the mean of non-sequential order correlations (Fig 5) indicating that the opinion change during the deliberative polling did not reflect a

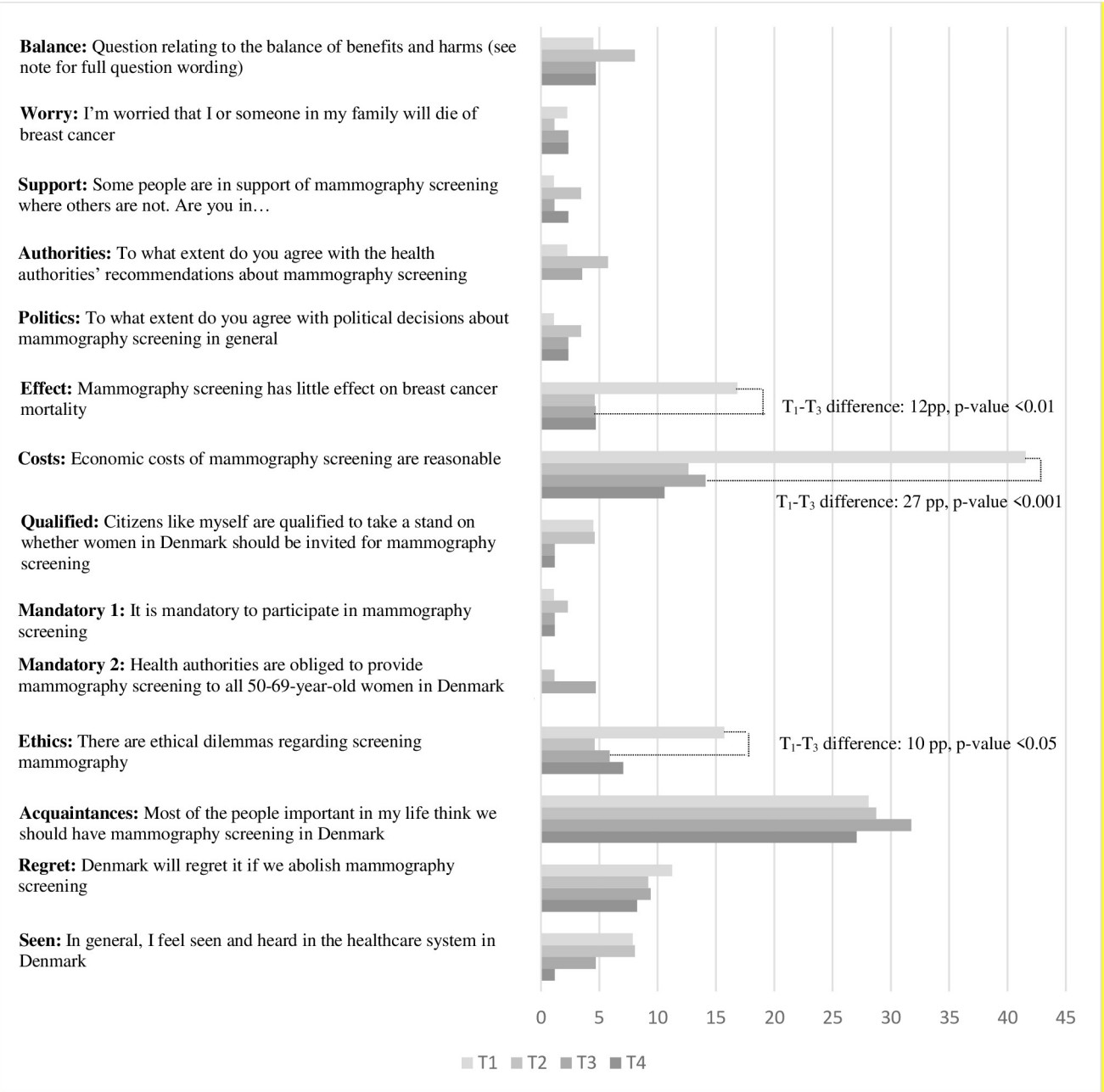

**Fig 4. Opinions formation—% 'Don't knows'.** Note: The figure shows the percentage of participants not able to form an opinion (answering 'don't know') to each of the 14 opinion items.

random process. These results support our second hypothesis: the process of information and deliberation lead to more stable opinions.

## 3. Opinion consistency

Consistency was operationalised as correlations *between* the 14 opinion items *within* each timepoint based on Converse's ideas [42]. Even though correlations are not measuring consistency but covariance, they are used as a measure for consistency as they specify whether variation in one variable corresponds to variation in another [45].

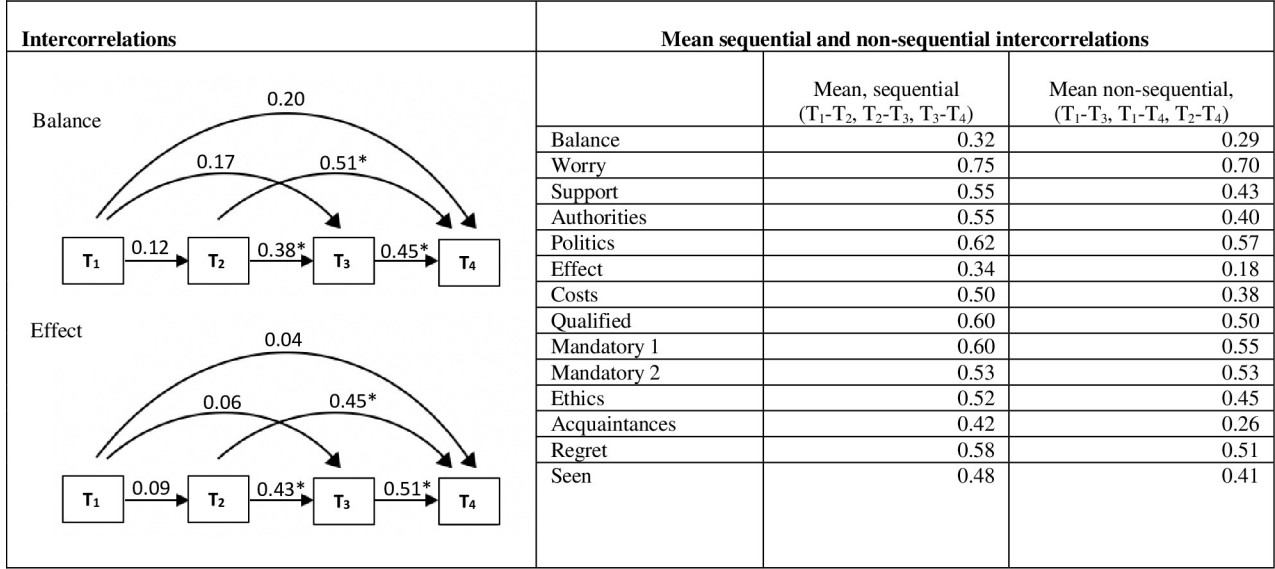

**Fig 5. Stability—Intercorrelations within opinions over time.** Note: The figure shows the intercorrelations within the opinions "Balance" and "Effect" over time. (Intercorrelations for the remaining opinion items are shown in S3 Fig) The figure shows the mean sequential order correlations as well as non-sequential order correlations. The more a correlation approximates one the more the answers from the two timepoints are in line. $T_1$ = inquiry time point 1 (recruitment), $T_2$ = inquiry time point 2 (after video information), $T_3$ = inquiry time point 3 (after deliberation), $T_4$ = inquiry time point 4 (one month after the citizens' assembly). See Fig 4 for full question wording.

In all correlation analyses 'don't know' answers were excluded. The number of statistically significant between-item correlations increased from 40 at recruitment ($T_1$) to 46 after the process of information and deliberation ($T_3$) (Figs 6 and S4). Of the 33 between-item correlations that were statistically significant at both $T_1$ and $T_3$, 22 increased from $T_1$ to $T_3$, ten decreased and one remained the same. Three between-item correlations deemed theoretically related a priori increased from $T_1$ to $T_3$: Support-Balance (from -0.28 to -0.55); Support-Effect (from -0.37 to -0.44), and Support-Costs (from 0.31 to 0.49). These results support our third hypothesis: the process of information and deliberation increased consistency between items deemed theoretically related, reflecting that participants had more clarity about their views.

## Discussion

To our knowledge we conducted the first Deliberative Poll on mammography screening and therefore our study is the first to report on laypeople's decision quality regarding mammography screening using this method. The method however have been widely used within other areas [46,47] and assessment of opinion quality using this method have been done before [43].

Our study revealed that it is possible for laypeople to form considered and informed opinions about mammography screening based on a process of video information and online deliberation. It might be supposed that citizens could be in greater doubt about their opinions when engaging on such a complex issue, but we found that more people formed opinions after information and deliberation compared to before. At the moment, the public is not in a position to form reasoned opinions about screening as baseline knowledge is low, which has been shown in several other studies [48–50]. In line with our expectations, laypeople were able to gain complex knowledge about mammography screening. Conceptual knowledge increased relatively more than numerical knowledge, even though the definition of numerical knowledge permitted a relatively wide range of correct answers (S5 Table). Despite efforts to aid numerical learning (rounded numbers and provision of a fact sheet with numbers), numerical

knowledge gain was relatively lower (21–33%) after information and deliberation (S5 Table), but still substantially increased compared to time of recruitment. Conceptual knowledge, on the other hand, was high after information and deliberation where more than 80% gave correct responses to seven out of nine items.

In line with our expectations, more people formed opinions during the deliberative process as more took a stand on screening effect size, costs and ethical dilemmas in screening (Fig 4, S6 Table). Moreover, and surprisingly, people in general expressed a formed opinion on most items at recruitment ($T_1$), as few answered 'don't know' to the questions. It could simply be that people already do have views on the topic. An alternative interpretation could be that high levels of media attention towards preventive strategies generate 'top-of-mind' opinions, or that people do not like to admit that they 'don't know'. The latter has been demonstrated in studies, where people respond with apparent opinions on named legislation that does not exist [10]. However, the questionnaires in the present study began with text addressing the possibility of answering 'don't know' as an attempt to comply with the described phenomenon. As no increase in 'don't know' responses was seen, participants did not become more doubtful of their opinions.

Participants, among whom the majority favoured screening at recruitment, weighted competing arguments during deliberation (Tables 4 and S3), and 89% said that all opinions were listened to with the same respect while 67% said that they gained an understanding of other participants' arguments that contradicted their own. Opinion stability increased (Figs 5 and S3) and consistency between theoretically related opinion items also increased (Figs 6 and S4) indicating opinion change related to the deliberative process and not a random change.

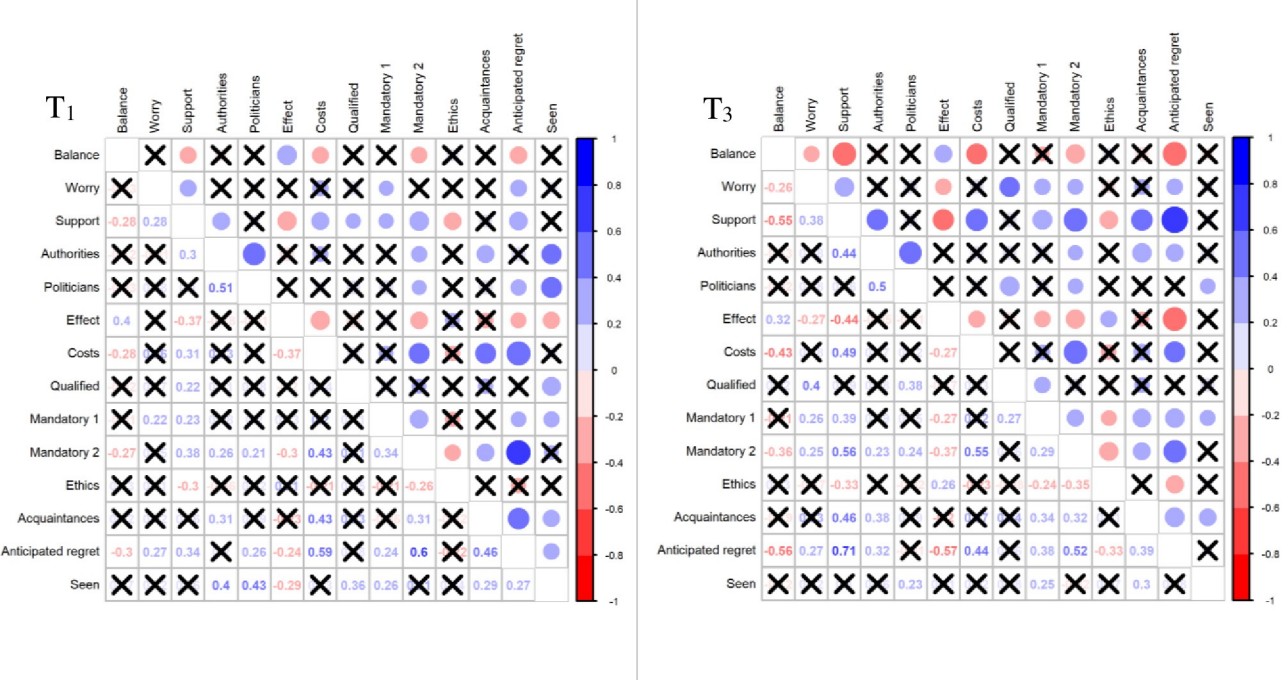

**Fig 6. Opinion consistency.** Note: The figure shows the internal correlation between opinion items at two time points ($T_1$ and $T_3$). A correlation of 1.00 (or -1) indicates a perfectly consistent relationship. Circle size indicates the size of statistically significant correlations. Blue colour when positively correlated, red colour when negatively correlated. X indicates no statistically significant correlation. $T_1$ = inquiry time point 1 (recruitment), $T_3$ = inquiry time point 3 (after deliberation). See Fig 4 for full question wording.

33% of participants felt there was too little time to discuss in group sessions whereas 34% felt there was enough time (S3 Table). More time for deliberation might have helped participants in doubt (answering 'don't know').

Participants evaluated the video positively: 71% did not feel that the video pushed their opinion in a certain direction whereas 29% felt that it did (S3 Table). We do not know how many of those 29% viewed the video as slanting away from versus slanting towards mammography screening. Looking at findings from a randomised controlled trial comparing two decision aids about mammography screening, one that contained information on overdetection and one that did not, it could be both [12]. It might be that when a population already enthusiastic about screening is informed about harms, the new information that conflicts with their main beliefs is viewed as pulling opinions in the opposite direction. Overall, participants in the Deliberative Poll evaluated the video information very positively: 96% reported that the video added new insights on mammography screening; 76% reported that it made their opinions more nuanced; 80% felt prepared to form opinions about mammography screening based on the video, and 69% would recommend it to peers (S3 Table).

The video had the greatest influence on participants' knowledge as seen in Fig 3. However, the majority of participants (68%) reported plenary discussions with experts as having the greatest impact on their opinions. The combination of video information and deliberation at the citizens' assembly seems to complement each other well. Overall, the study demonstrated that it is possible to gather a representative group of people in meaningful online deliberation about mammography screening, and 85% of the study participants reported that they would recommend participating in a citizens' assembly to peers.

We acknowledge that our results should be viewed in light of the evidence we provided to participants, how we conceptualised decision quality as well as our deliberative poll setup including time for deliberation.

## Strength and limitations

An advantage of this study was the rigorous way the design ensured close to perfect sociodemographic as well as attitudinal representativeness. Another advantage relates to the repeated measurement design. This longitudinal design with four waves of items addressed to the same participants ensured statistical power without an even larger sample size. In addition, the design allowed for analysis over time. Both women (with and without a history of breast cancer) and men were included in the Deliberative Poll. This was to gain insights into the general public's perspectives about mammography screening in the light of the involved trade-offs when providing a healthcare service. This is in contrast to most citizens' juries which tend to include only women without a history of breast cancer [33–35,51]. Compared to a physical assembly, the online format brought cost savings related to travel, venue and catering. Our findings should be interpreted in light of two limitations.

First, selection bias. The initial response rate was 63%. As with all studies where participation is not mandatory, some selection bias is inevitable. Attempts made to minimize selection bias were the provision of an online rehearsal with the purpose of familiarising the participants with the online platform and their technical equipment at home. Around 40% took part in one of the two rehearsals held in the days before the assembly. The two-step quota sampling process meant that the response rate below 100% did not affect representativeness on the known sociodemographic and attitudinal parameters.

Second, retest bias. We cannot exclude the possibility that repeated measurement in this study in itself leads to an increase in performance on knowledge items [52]. An equivalent phenomenon related to opinion formation is the Socratic effect—where questions in themselves

can cause the respondent to reflect on their opinions and behaviour, leading to increased consistency [41,53]. Inclusion of a control group tested only after viewing the video could have been beneficial to assess any potential retest effect. However, as the impact of the study intervention on knowledge was large, we believe that most of the increased knowledge resulted from actual learning and not a retest phenomenon.

Our study shows that citizens' engagement through Deliberative Polling improves their knowledge, opinion consistency, stability and decision-making ability on the very complex issue of balancing benefits and harms in mammography screening.

## Implications

This study provides much optimism around how to engage with public opinion in the future concerning complex health programmes. To our knowledge, this is the first study to demonstrate that it is possible to involve a representative group of people at a national level in online deliberation. In light of the ongoing COVID-19 pandemic, this is an important finding as physical assemblies are currently prohibited in many countries and may remain an area of uncertainty in the years to come. The online format is also cost and time saving, and environmentally more sustainable, giving it great potential in future deliberative events. If there is a need for engagement of the public, using an online Deliberative Poll is both feasible and cheap.

It is the politicians who decide if a screening programme should be implemented based on recommendations made by experts. Recommendations from the deliberative poll could be used to inform this policy. Engaging with the public have the potential to increase the chance of a successful implementation or termination of screening programmes due to increased understanding of the different positions and reasons underlying the decision.

## Supporting information

**S1 Fig. Factsheet (translated from Danish).**
(TIF)

**S2 Fig. Questionnaire $T_3$ (Danish).**
(TIF)

**S3 Fig. Stability, intercorrelations within opinions over time.** Note: The figure shows the intercorrelations within the opinions over time. The more a correlation approximates one the more the answers from the two timepoints are in line. n varies: between 89 and 85. $T_1$ = inquiry time point 1 (recruitment), $T_2$ = inquiry time point 2 (after video information), $T_3$ = inquiry time point 3 (after deliberation), $T_4$ = inquiry time point 4 (one month after the citizens' assembly).
(TIF)

**S4 Fig. Consistency, internal correlation *between* opinion items at the four time points.** Note: The figure shows the internal correlation *between* opinion items at four time points. A correlation of 1.00 (or -1) indicates a perfectly consistent relationship. Circle size indicates the size of statistically significant correlations. Blue color when positively correlated, red color when negatively correlated. X indicates no statistically significant correlation. $T_1$ = inquiry time point 1 (recruitment), $T_2$ = inquiry time point 2 (after video information), $T_3$ = inquiry time point 3 (after deliberation), $T_4$ = inquiry time point 4 (one month after the citizens' assembly).
(TIF)

**S1 Table. Program, citizens' assembly.**
(TIF)

**S2 Table. Description of video information and questionnaire development.**
(TIF)

**S3 Table. Participants evaluation of the Deliberative Poll.** Note: This table shows participants' evaluation of group discussions, moderators, video information and the Deliberative Poll overall. $T_2$ = inquiry time point 2 (after video information), $T_3$ = inquiry time point 3 (after deliberation).
(TIF)

**S4 Table. Mean level of knowledge (index 0–100) divided according to sociodemographic characteristics as well as worry and knowledge starting point.** Note: The knowledge index combines all 13 questions giving 7.69 points for each correct answer. The index ranges from 0 to 100 where 100 indicates correct answers to all questions and 0 incorrect answers to all questions. n varies between timepoints. $T_1$ = inquiry time point 1 (recruitment), $T_2$ = inquiry time point 2 (after video information), $T_3$ = inquiry time point 3 (after deliberation), $T_4$ = inquiry time point 4 (one month after the citizens' assembly).
(TIF)

**S5 Table. Level of knowledge (% correct answers).** Note: The table shows the level of knowledge at the four poll inquiry time points expressed as % correct answers to the 13 knowledge items. $T_1$ = inquiry time point 1 (recruitment), $T_2$ = inquiry time point 2 (after video information), $T_3$ = inquiry time point 3 (after deliberation), $T_4$ = inquiry time point 4 (one month after the citizens' assembly).
(TIF)

**S6 Table. Opinion formation (% 'Don't knows').** Note: The table show the percentage of participants not able to form an opinion (answering 'don't know') to each of the 14 opinion items. $T_1$ = inquiry time point 1 (recruitment), $T_2$ = inquiry time point 2 (after video information), $T_3$ = inquiry time point 3 (after deliberation), $T_4$ = inquiry time point 4 (one month after the citizens' assembly).
(TIF)

**S7 Table. Reference list for supporting information.**
(TIF)

## Acknowledgments

We would like to thank the following people for helping us to complete this project: all the participants; the three experts who contributed at the assembly and the moderators of the group discussions. Finally, we wish to extend a special thanks to the chair of the online citizens' assembly Ole Hartling.

## Author Contributions

**Conceptualization:** Manja D. Jensen, Kasper M. Hansen, Volkert Siersma, John Brodersen.

**Formal analysis:** Manja D. Jensen, Volkert Siersma.

**Funding acquisition:** Manja D. Jensen, John Brodersen.

**Investigation:** Manja D. Jensen, Kasper M. Hansen, John Brodersen.

**Methodology:** Kasper M. Hansen.

**Project administration:** Manja D. Jensen.

**Supervision:** Kasper M. Hansen, Volkert Siersma, John Brodersen.

**Validation:** Volkert Siersma.

**Visualization:** Manja D. Jensen, Kasper M. Hansen, Volkert Siersma, John Brodersen.

**Writing – original draft:** Manja D. Jensen.

**Writing – review & editing:** Manja D. Jensen, Kasper M. Hansen, Volkert Siersma, John Brodersen.

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
