## [Decision Letter · Decision Letter 0]

20 Sep 2021

PONE-D-21-26133In search of good advice: How deliberation on breast cancer screening improves the decision quality of laypeoplePLOS ONE

Dear Dr. Jensen,

Thank you for submitting your manuscript to PLOS ONE. After careful consideration, we feel that it has merit but does not fully meet PLOS ONE’s publication criteria as it currently stands. Therefore, we invite you to submit a revised version of the manuscript that addresses the points raised during the review process.

The feedback from reviewers and academic editor are detailed below. The reviewers felt this required minor revisions, which I agree with. I've extended the scope of revisions but these are not required, rather for consideration of the authors to expand the reproducibility and impact, keeping in mind that there are no restrictions on length with PLoS One. Any data or additinoal tables that could be included as a supplementary are also worth considering and of value.

We look forward to receiving your revised manuscript.

Kind regards,

Dylan A Mordaunt

Academic Editor

PLOS ONE

Additional Editor Comments:

Thank you for your submission. With regards to criteria for publication: 1)The study presents the results of original research, 2) Results do not appear to have been published elsewhere, 3) Experiments, statistics, and other analyses are performed to a high technical standard and are largely described in sufficient detail, with some minor amendments needed. 4) Conclusions are presented in an appropriate fashion and are supported by the data, 5) The article is presented in an intelligible fashion and is written in standard English, 6) The research meets all applicable standards for the ethics of experimentation and research integrity, 7) The research meets all applicable standards for the ethics of experimentation and research integrity. The authors have provided some valuable feedback. It would be worth ensuring that there is some epistemological context, methods fully described, discussion clearly referencing relevant literature and expansion on the valuable section on implications to include how this is of relevance to application in program and policy formulation.

Journal Requirements:

4. Please upload a new copy of Figure 6 as the detail is not clear. Please follow the link for more information: https://blogs.plos.org/plos/2019/06/looking-good-tips-for-creating-your-plos-figures-graphics/

Reviewers' comments:

Reviewer's Responses to Questions

**Comments to the Author**

1. Is the manuscript technically sound, and do the data support the conclusions?

Reviewer #1: Yes

Reviewer #2: Yes

2. Has the statistical analysis been performed appropriately and rigorously? 

Reviewer #1: Yes

Reviewer #2: I Don't Know

3. Have the authors made all data underlying the findings in their manuscript fully available?

Reviewer #1: Yes

Reviewer #2: No

4. Is the manuscript presented in an intelligible fashion and written in standard English?

Reviewer #1: Yes

Reviewer #2: Yes

5. Review Comments to the Author

Reviewer #1: This is a well written paper, describing the role of decision aids like deliberation on breast cancer screening improves the decision quality of laypeople. This paper adds to the current knowledge about the current decision abilities for mammogram screening.

1) Can the authors specify how the study challenges existing paradigms

2) Please mention if there has been previous work in the same space? if not highlight what is relevant/novel about this hypothesis

3) Did the subjects get enough time to deliberate carefully about the option to choose screening ?

Reviewer #2: - The Full Title and the Short Title do not agree. I suggest removing the "In search of good advice" from the Full title and adding the Short title (with the words "Use of") at the end of it after a colon. Then the Short title could be "Use of a Deliberative Poll...."

- The definition of a Deliberative Poll should be done earlier, perhaps in the Abstract and then repeated in the Deliberative Poll section.

- In the "Judging the quality of..." section at the end of the fifth paragraph in the sentence "...are participants able to form....", it seems that the word should be "decision" instead of "advice", unless one is assuming that the knowledge participants gained would be used to advise others.

- The process of recruitment of participants depended on telephone interviews. There is no discussion of whether that process could be biased because fewer people have landlines than cell phones at present. The question of who has a landline vs. who has only a cell phone needs to be addressed.

- I applaud the use of comparisons between respondents and the Danish population in general in Table 1.

- Although the Citizens' assembly was conducted online as opposed to in a physical location, making it easier for participants, it still required a full day. The possibility of Selection Bias is high, since participants who work full-time or take care of their children during the day would exclude their participation. Was this accounted for?

- The Discussion section included many important points, especially the remark that people don't like to admit that they "don't know" something and that may have influenced answered to the questionnaire.

6. PLOS authors have the option to publish the peer review history of their article (what does this mean?). If published, this will include your full peer review and any attached files.

Reviewer #1: No

Reviewer #2: No

---

## [Author Response · Author response to Decision Letter 0]

4 Oct 2021

Dear Reviewers

Thank you for your insightful comments and suggestions which have helped us improve the manuscript. Below we respond to each of your comments in turn and show how we have changed the manuscript in order to take them into account. All changes are furthermore marked by track changes in the revised manuscript.

Reviewer #1: 

1) Can the authors specify how the study challenges existing paradigms. 

Thank you for the comment. Today the public opinion about screening is not formally considered in the decision-making process about screening programmes. This is now described in the section “implications” in the manuscript marked “Revised Manuscript marked with Track Changes”

2) Please mention if there has been previous work in the same space? if not highlight what is relevant/novel about this hypothesis. 

Thank you for the comment. To our knowledge we conducted the first Deliberative Poll on mammography screening. This is now described in the beginning of the section “Discussion”.

3) Did the subjects get enough time to deliberate carefully about the option to choose screening?

Thank you for this question. 2/3 disregarded “There was too little time to discuss” (S3 Table). In the revised manuscript we have included this in the discussion. 

Reviewer #2: 

1) The Full Title and the Short Title do not agree. I suggest removing the "In search of good advice" from the Full title and adding the Short title (with the words "Use of") at the end of it after a colon. Then the Short title could be "Use of a Deliberative Poll...."

Thanks for the comment. We have rephrased the title. The changes appear in the manuscript marked “Revised Manuscript marked with Track Changes”

2) The definition of a Deliberative Poll should be done earlier, perhaps in the Abstract and then repeated in the Deliberative Poll section. 

Thanks for the comment. We have added the definition to the abstract section. Now the definition appears in the abstract section as well as in the section “The Deliberative Poll – Deliberation and representativeness”.

3) In the "Judging the quality of..." section at the end of the fifth paragraph in the sentence "...are participants able to form....", it seems that the word should be "decision" instead of "advice", unless one is assuming that the knowledge participants gained would be used to advise others. 

Thanks for the comment. We have changed the word “advice” to “decision”.

4) The process of recruitment of participants depended on telephone interviews. There is no discussion of whether that process could be biased because fewer people have landlines than cell phones at present. The question of who has a landline vs. who has only a cell phone needs to be addressed. 

Thanks for the comment. Kantar Gallup’s online panel of 50.000 citizens were recruited through random sampled telephone interviews. Citizens were recruited either by their landline or their mobile phone dependent on which was obtainable. We have added this information in the section “recruitment”.

5) I applaud the use of comparisons between respondents and the Danish population in general in Table 1. 

Thank you.

6) Although the Citizens' assembly was conducted online as opposed to in a physical location, making it easier for participants, it still required a full day. The possibility of Selection Bias is high, since participants who work full-time or take care of their children during the day would exclude their participation. Was this accounted for? Thank you for the comment. The Deliberative Poll was conducted on a Sunday. We specifically chose this day of the week to reduce selection bias related to participants going to work. In addition, we compensated participants economically, for example to pay for babysitting or taking a day off (if working on a Sunday). We have elaborated on this in the section “The Deliberative Poll on mammography screening”.

7) The Discussion section included many important points, especially the remark that people don't like to admit that they "don't know" something and that may have influenced answered to the questionnaire. 

Thank you.

Additional Editor Comments:

Thank you for your submission. With regards to criteria for publication: 1) The study presents the results of original research, 2) Results do not appear to have been published elsewhere, 3) Experiments, statistics, and other analyses are performed to a high technical standard and are largely described in sufficient detail, with some minor amendments needed. 4) Conclusions are presented in an appropriate fashion and are supported by the data, 5) The article is presented in an intelligible fashion and is written in standard English, 6) The research meets all applicable standards for the ethics of experimentation and research integrity, 7) The research meets all applicable standards for the ethics of experimentation and research integrity. The authors have provided some valuable feedback. It would be worth ensuring that there is some epistemological context, methods fully described, discussion clearly referencing relevant literature and expansion on the valuable section on implications to include how this is of relevance to application in program and policy formulation. 

Thank you for the comments. In the revised manuscript we have elaborated on how participants made an informed consent in the method section “Recruitment”. In addition, we have added information regarding the decision by the Committees on Health Research Ethics. They did not regard our study as a health research project, but considered it a questionnaire-based study and therefore it was not subject to notification from the Committees. The decision is filed as Journal-no.: 21031705. With regard to the references we have removed the reference “Wilson J, Jungner G. Principles and practice of screening for disease. 1968. Public Health Papers” and replaced it with the following: “WHO report: Screening programmes, a short guide 2020.” In addition we have referenced “Center for Deliberative Democracy: CDD - Stanford University”, “Hansen KM. Deliberative Democracy and Opinion Formation” and “Degeling C, Carter SM, Rychetnik L. Which public and why deliberate? – A scoping review of public deliberation in public health and health policy research” in the discussion section. Misspelling is corrected in the reference list regarding “Baena-Cañada JM, Luque-Ribelles V, Quílez-Cutillas A, Rosado-Varela P, Benítez-Rodríguez E, Márquez-Calderón S, et al. How a deliberative approach includes women in the decisions of screening mammography: a citizens'; jury feasibility study in Andalusia, Spain. BMJ open. 2018;8(5):e019852. doi: 10.1136/bmjopen-2017-019852.” 

We have made an addition to the implications section and stated in the discussion section that we acknowledge that our results should be viewed in light of the evidence we provided to participants, how we conceptualised decision quality as well as our deliberative poll setup including time for deliberation.

Journal Requirements:

Thanks for the comments. The title page has been revised to comply with style requirements. The file name of all the supplementary files are now changes from S(number).Fig.tif to S(number)_Fig.tif. 

2) We note that you have indicated that data from this study are available upon request. PLOS only allows data to be available upon request if there are legal or ethical restrictions on sharing data publicly. For more information on unacceptable data access restrictions, please see http://journals.plos.org/plosone/s/data-availability#loc-unacceptable-data-access-restrictions. There are no ethical or legal restrictions on sharing our anonymized data set. We have uploaded our anonymized datasets and our SAS-file necessary to replicate our findings at Harvard Dataverse:

https://dataverse.harvard.edu/dataverse/MKVJ. The SAS code and the three SAS dataset can also be assessed directly:

https://doi.org/10.7910/DVN/ZQHO5W

https://doi.org/10.7910/DVN/WFJIUV

https://doi.org/10.7910/DVN/SSNZJL

https://doi.org/10.7910/DVN/DYFUS9

3) Please include your full ethics statement in the ‘Methods’ section of your manuscript file. In your statement, please include the full name of the IRB or ethics committee who approved or waived your study, as well as whether or not you obtained informed written or verbal consent. If consent was waived for your study, please include this information in your statement as well. We have included the full ethics statement in the methods section and added the Danish name of the ethics committee. In addition, we have described how we obtained informed consent in more detail.

4) Please upload a new copy of Figure 6 as the detail is not clear. Please follow the link for more information: https://blogs.plos.org/plos/2019/06/looking-good-tips-for-creating-your-plos-figures-graphics/

We have uploaded a new copy of figure 6 (and S4_Fig). 

To make sure that the details stay clear we have uploaded the different parts of the figures separately: 

•For Fig.6 we have uploaded: Fig.6.T1 and Fig.6.T3. 

•For S4_Fig we have uploaded: S4_FigT1 and S4_FigT2 and S4_FigT3 and S4_FigT4. 

If we misunderstand what you mean by “clear” please let us know and we will look at the figure layout again. We have uploaded a new copy of figure 5 as well because of misspelling in the figure. 

5) Please review your reference list to ensure that it is complete and correct. If you have cited papers that have been retracted, please include the rationale for doing so in the manuscript text, or remove these references and replace them with relevant current references. Any changes to the reference list should be mentioned in the rebuttal letter that accompanies your revised manuscript. If you need to cite a retracted article, indicate the article’s retracted status in the References list and also include a citation and full reference for the retraction notice

Thanks for the comment. We have reviewed the reference list and removed the reference “Wilson J, Jungner G. Principles and practice of screening for disease. 1968. Public Health Papers. 1968;34.” The reference is replaced with the following: “WHO report: Screening programmes, a short guide 2020. Available from: https://apps.who.int/iris/bitstream/handle/10665/330829/9789289054782-eng.pdf

---

## [Decision Letter · Decision Letter 1]

7 Oct 2021

Using a Deliberative Poll on breast cancer screening to assess and improve the decision quality of laypeople

PONE-D-21-26133R1

Dear Dr. Jensen,

We’re pleased to inform you that your manuscript has been judged scientifically suitable for publication and will be formally accepted for publication once it meets all outstanding technical requirements.

Kind regards,

Dylan A Mordaunt, MB ChB, FRACP, FAIDH

Academic Editor

PLOS ONE

Additional Editor Comments (optional):

Reviewers' comments:

Reviewer's Responses to Questions

**Comments to the Author**

1. If the authors have adequately addressed your comments raised in a previous round of review and you feel that this manuscript is now acceptable for publication, you may indicate that here to bypass the “Comments to the Author” section, enter your conflict of interest statement in the “Confidential to Editor” section, and submit your "Accept" recommendation.

Reviewer #1: All comments have been addressed

Reviewer #2: All comments have been addressed

2. Is the manuscript technically sound, and do the data support the conclusions?

Reviewer #1: Yes

Reviewer #2: (No Response)

3. Has the statistical analysis been performed appropriately and rigorously? 

Reviewer #1: Yes

Reviewer #2: (No Response)

4. Have the authors made all data underlying the findings in their manuscript fully available?

Reviewer #1: Yes

Reviewer #2: (No Response)

5. Is the manuscript presented in an intelligible fashion and written in standard English?

Reviewer #1: Yes

Reviewer #2: (No Response)

6. Review Comments to the Author

Reviewer #1: Thank you for the suggested changes, the revision has strengthened the paper. The manuscript is important and relevant to current times.

Reviewer #2: (No Response)

7. PLOS authors have the option to publish the peer review history of their article (what does this mean?). If published, this will include your full peer review and any attached files.

Reviewer #1: No

Reviewer #2: No

---

## [Editor Report · Acceptance letter]

12 Oct 2021

PONE-D-21-26133R1 

Using a Deliberative Poll on breast cancer screening to assess and improve the decision quality of laypeople 

Dear Dr. Jensen:

I'm pleased to inform you that your manuscript has been deemed suitable for publication in PLOS ONE. Congratulations! Your manuscript is now with our production department. 

Kind regards, 

on behalf of

Dr. Dylan A Mordaunt 

Academic Editor

PLOS ONE